# Double-Resolved Beam Steering by Metagrating-Based Tamm Plasmon Polariton

**DOI:** 10.3390/ma15176014

**Published:** 2022-08-31

**Authors:** Rashid G. Bikbaev, Dmitrii N. Maksimov, Kuo-Ping Chen, Ivan V. Timofeev

**Affiliations:** 1Kirensky Institute of Physics, Federal Research Center KSC SB RAS, Krasnoyarsk 660036, Russia; 2Siberian Federal University, Krasnoyarsk 660041, Russia; 3Institute of Photonics Technologies, National Tsing Hua University, Hsinchu 30013, Taiwan; 4Institute of Imaging and Biomedical Photonics, National Yang Ming Chiao Tung University, Tainan 71150, Taiwan

**Keywords:** metasurface, metagratings, tamm plasmon polaritons, chirality

## Abstract

We consider Tamm plasmon polariton in a subwavelength grating patterned on top of a Bragg reflector. We demonstrate dynamic control of the phase and amplitude of a plane wave reflected from such metagrating due to resonant coupling with the Tamm plasmon polariton. The tunability of the phase and amplitude of the reflected wave arises from modulation of the refractive index of a transparent conductive oxide layer by applying the bias voltage. The electrical switching of diffracted beams of the ±1st order is shown. The possibility of doubling the angular resolution of beam steering by using asymmetric reflected phase distribution with integer and half-integer periods of the metagrating is demonstrated.

## 1. Introduction

The metasurface is an artificially structured layer with subwavelength-scaled elements designed to effectively control the phase and amplitude of reflected light [1,2]. The interest in such systems is fueled by a significant advance in subwavelength technologies such as optical and e-beam lithography. Metasurfaces open up new opportunities for implementation of holograms [3], lenses [4], media with anomalous reflection [5,6], lasers [7,8], perfect absorber with critical coupling [9], etc. Structures in which meta-atoms or metamolecules form two-dimensional periodic lattices are often called metagratings [10,11]. Such constructions have the ability to direct light into a certain diffraction channel, achieving remarkable efficiency even at very large bending angles [6,12]. An actively tuned metasurface and metagrating with control of phase and amplitude of individual elements ensures the generation of an arbitrarily complex wave front. There are several approaches to actively rearrange the optical properties of a metasurface and metagrating. The first one is application of liquid crystals [13,14,15]. The advantage of this method is the control over optical properties via variation of both electric field and temperature. However, one of the key drawbacks of such devices is their large switching times of several milliseconds. Another promising approach is application of phase-change metasurfaces and metagrating. These systems have proven to have excellent switching times of microseconds, but the phase modulation schemes applied so far only allowed for discrete switching [16]. Variation of the external electric field has already been implemented in plasmonic amplitude modulators [17,18]. It is shown that classical semiconductor materials can be replaced by transparent conducting oxides. Thus, the effective control of the volume concentration of charge carriers at the interface between a conducting oxide and a metal film was demonstrated [19,20].

It was shown that the increase in the applied bias voltage leads to a zero and even negative real part of the permittivity of the conducting oxide in a very thin layer near the metal. This results in a significant field enhancement in this region and provides the necessary phase jump near the resonant wavelength.

In this work, we demonstrate effective tuning of the wavefront in the near infrared region by introducing a controlled material into a structure (Figure 1a) supporting Tamm plasmon polaritons (TTP) [21,22,23,24,25]. The beam steering in indium tin oxide (ITO) [26,27] metasurfaces has been previously demonstrated in [20]. Here we show that our scheme allows for doubling the angular resolution of beam steering with phase distributions with integer and half-integer periods of the metagrating. The advantage of the proposed TTP-based structure compared to conventional metal–insulator–metal devices is the excitation of resonance with a higher Q-factor which makes the phase more sensitive to the control parameters. Moreover, the proposed structure is attractive for implementing photonic crystal surface-emitting lasers (PCSEL) [28]. Thus, the setup proposed paves the way for engineering optical devices with both PCSEL and beam steering functionality.

## 2. Model

### 2.1. Electrostatic Simulation

The electron distribution in the ITO layer as a function of applied bias voltage was define by numerically solving the Poisson and drift-diffusion equations. The ITO has been presented as a semiconductor with the bandgap of E=2.8 eV, electron affinity of χ=5 eV, effective electron mass of $m^{*}=0.25m_{e}$ and permittivity ε=9. The ITO carrier concentration is $N_{0}$ = 2.8 × 10${}^{20}$ cm${}^{-3}$. The DC permittivity of the Al${}_{2}$O${}_{3}$ is $\varepsilon_{{\mathrm{Al}}_{2}O_{3}}=9$. In our simulation, the Ag nanostripes and a monolayer graphene were used as electric contacts. The change in the dielectric constant of Ag under the bias voltage was not taken into account. We used the mesh size of 0.1 nm which has been validated by performing careful convergence tests.

It can be seen from Figure 1b that at zero voltage, electron depletion is observed at the ITO-Al${}_{2}$O${}_{3}$ boundary. This is due to the fact that the working function of the ITO (4.4 eV) at this concentration is lower than the working function of silver (4.73 eV). The increase of the bias voltage between nanostripes and graphene layer leads to electron accumulation at the ITO-Al${}_{2}$O${}_{3}$ interface. Figure 1c shows the dependence of the real part of the complex permittivity of the ITO layer at a wavelength of 1550 nm on the applied bias voltage and the distance from the Al${}_{2}$O${}_{3}$ boundary. The increase in voltage leads to a significant change in the dielectric constant of the conductive oxide in a thin 3 nm layer. At voltages greater than 2 V, it takes values close to zero. The epsilon near zero region area is highlighted with blue. A further increase in voltage leads to the real part of the dielectric constant becoming negative.

### 2.2. Full-Wave Simulation

The reflection spectra of the structure from Figure 1a and the reflection phases for normally incident TM-polarized waves were calculated by the finite-difference time domain (FDTD) method. The DBR unit cell is formed by silica and titanium dioxides. The geometric parameters of the DBR are listed in the caption of Figure 1a.

The reflectance spectra of the structure at different values of the bias voltage applied to the ITO are shown in Figure 2a. The dip observed near λ=1550 nm in Figure 2a corresponds to the Tamm plasmon polariton localized at the interface between the metagrating and the multilayer mirror [29]. The minimum reflection at the TPP wavelength is due to the critical couplin, at which the rate of energy decay into the radiation and absorption channels of the metagrating are comparable. In practice, this effect is achieved by increasing the number of layers of the multilayer mirror and tuning the height of silver nanostrips. An increase in the bias voltage leads to a blue shift of the resonant wavelength. This effect can be explained by the fact that the increase in voltage leads to an increase in the volume concentration of the charge carriers near the ITO-Al${}_{2}$O${}_{3}$ boundary. As a result, the real part of the complex dielectric permittivity of the ITO becomes negative, and it acquires metallic properties. Thus, an additional term appears in the expression of phase matching corresponding to the phase winding when passing through the ITO layer. The blue shift of the localized state leads to a significant change in the reflection phase at a wavelength of λ=1550 nm (see Figure 2b). Thus, at a voltage of 3 V, it is possible to achieve a phase change of 180${}^{\circ }$, while at the maximum considered voltage of 5 V, the phase is 214${}^{\circ }$.

The calculations have shown that by changing the bias voltage applied to the ITO film, it is possible to control the reflection phase of each nanostrip. Thus, it becomes possible to create a tunable diffraction grating, the period of which is determined by the number of nanostrips with different applied bias voltages. The angular resolution of such a device is limited by the width of the unit cell, since switching is carried out discretely, see the diagram in Figure 2c. So, in the case when a voltage of 0 V is applied to two strips, and 3.5 V to the next two strips, the lattice period is two microns. Changing the number of strips from four to eight leads to an increase of the grating period to four microns and, as a consequence, results in a change of the first-order diffraction angle:(1)$$\Theta_{n}=-\arcsin\left(\frac{m\lambda }{np}\right)$$
here and after the diffraction order m=0,±1, and *n* is the number of strips used in the dynamic unit cell. In principle, *n* can be an arbitrary real number providing continuous steering rather than discrete steering for integer *n*. The drawback of non-integer *n* is that the TPPs of neighboring strips are mixed; their phases produce unrepeatable cell patterns that are hard to control. Nevertheless, rational *n* produces few different patterns that are periodic in space. For example, the half-integer value of *n* makes only two different cell patterns that lead to relatively sharp beams, as illustrated in Table 1. If three and more patterns are included, then the ±1 diffraction order intensity falls down and requires special inverse design of applied voltages for effective beam steering.

To numerically demonstrate this effect, we calculated the intensities of diffraction maxima in the far field. The calculation was carried out for 50 nanostrips. The results are presented in Figure 2d. It can be seen that a change in the lattice period leads to a significant (about 30${}^{\circ }$) change in the angles of −1 and +1 diffraction orders. Note that the intensity of zeroth order is equal to zero. This is explained by the destructive interference of waves propagating along the normal to the metagrating due to the fact that the phase difference of the reflected waves from neighboring nanostrips is equal to π.

We aim at doubling the angle resolution of the device by continuously tuning the voltage and, as a consequence, the phase from one nanostrip to another, see Figure 3a. For double resolution, one can add a diffraction grating period consisting of a half-integer number of nanostrips along with integer nanostrip periods, see Figure 3b. In this regard, it becomes relevant to solve the inverse problem, i.e., to calculate the phase distribution over nanostrips and to provide reflection at the necessary angle via the generalized Snell’s law:(2)$$\sin\left(\theta_{r}\right)-\sin\left(\theta_{i}\right)=\frac{\lambda }{2\pi }\frac{\Delta \Phi }{p},$$
where ΔΦ is the phase difference between neighbouring nanostrips, and $\theta_{r}$ and $\theta_{i}$ are reflection and incidence angles, respectively. To demonstrate this effect, the required phase distribution along the metagrating was determined, see Figure 3b. The calculations were performed for 50 nanostrips and for $\theta =38.3^{\circ }$ and $\theta =62.3^{\circ }$. It can be seen from Figure 3b that for an angle of 38.3${}^{\circ }$ the diffraction grating period is formed of five nanostrips, while the phase distribution turns out to be asymmetric. For larger angles, for example for 62.3${}^{\circ }$, the phase distribution is also asymmetric, but the period of the phase distribution consists of a half-integer number of nanostrips and is equal to 3.5, see Table 1.

Based on this distribution, the corresponding bias voltages were determined by interpolating the data presented in Figure 2b. According to the obtained results, the dependence of the refractive index of the ITO film on its thickness was determined. Then, the intensities of diffraction maxima in the far field were calculated. The results are shown in Figure 3b. It can be seen from Figure 3b that the intensity of −1 order is much greater than the intensity of +1 order. This is explained by the fact that the phase profile shown in Figure 3b leads to an increase in the intensity of the field only for −1 order, while for 0 and +1 order direction, the waves are destructively extinguished. Thus, the designed scheme allows one to control a nanoscale beam in a wide range of angles and has significant potential for engineering ultrathin devices such as LIDARs and nanoscale spatial light modulators.

## 3. Conclusions

The paper demonstrates the control of the phase and amplitude of a wave reflected from a Tamm plasmon polariton structure. The refractive index modulation in a thin layer of transparent conductive oxide located at the boundary of a multilayer mirror and a grating metagrating on the phase of the reflected wave is demonstrated. This effect makes it possible to use such a structure as a dynamic phase diffraction grating. The calculations showed that changing the number of nanostrips with different applied bias voltage allows for efficient switching in ± first diffraction orders. In addition, it is shown that the reflected beam can be controlled by a continuous phase change along the metagrating. This method allows us to increase the beam steering angular resolution.

## Figures and Tables

**Figure 1 materials-15-06014-f001:**
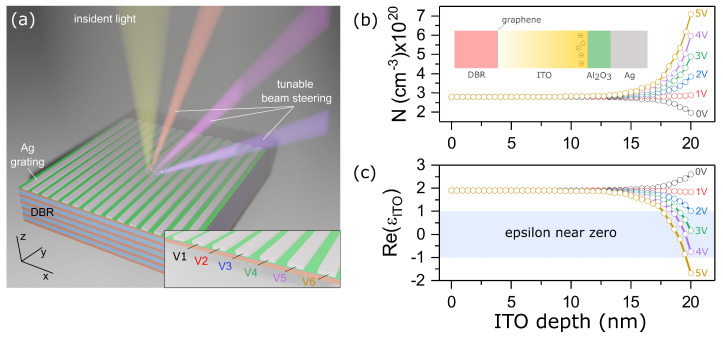
(**a**) Sketch view of the structure; (**b**,**c**) electron concentration *N* and real part of the dielectric permittivity ReεITO of the ITO layer for different applied bias voltage. The DBR layers’ thicknesses are $d_{a}=165$ nm and $d_{b}=135$ nm, for silica and titanium dioxide, correspondingly. The number of DBR layers are equal to 15. The 2D array with thickness h=95 nm and width L=470 nm has infinite length along the *y* axis. The pitch of the array p=500 nm. The ITO and Al${}_{2}$O${}_{3}$ layer thicknesses are 20 nm and 5 nm, respectively. The structure in the inset of Figure 1b is presented schematically to demonstrate the non-uniform distribution of charges in the ITO in the case of applying a bias voltage between Ag nanostripes and monolayer graphene.

**Figure 2 materials-15-06014-f002:**
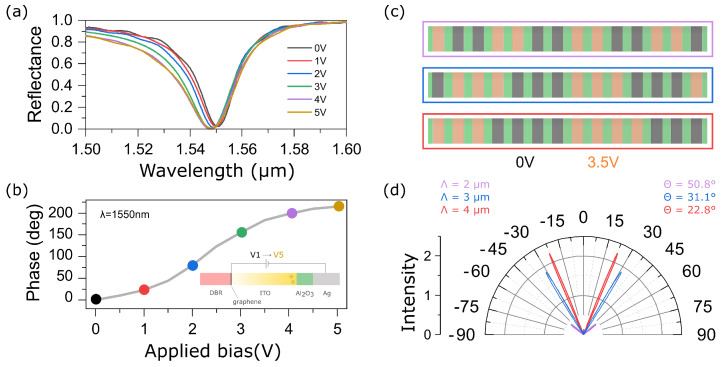
(**a**,**b**) Reflectance spectra of the structure presented in Figure 1a,b simulated phase shift as a function of applied bias voltage between Ag nanostripes and monolayer graphene; (**c**) schematic of the diffraction grating for a different number of nanostrip pairs; dark gray and orange nanostrips depict nanostrips without bias voltage and with bias voltage of 3.5 V, respectively; an increase in the number of nanostripes with bias voltage and without it leads to an increase in the grating period; (**d**) Simulated far-field reflected intensity from the metagrating as a function of the diffraction angles for a different grating period.

**Figure 3 materials-15-06014-f003:**
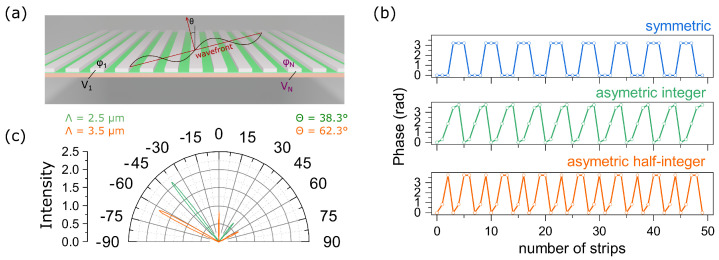
(**a**) Schematic representation of the structure for control over the angle of the first diffraction order; (**b**) three types of the phase distribution along the metagrating; (**c**) simulated far-field reflected intensity from the metagrating based on phase distribution presented in (**b**) in the case of asymmetric phase distribution.

**Table 1 materials-15-06014-t001:** The dependence of the ± first order diffraction angle on number of nanostrips. Half-integer *n* corresponds to double resolution.

Number of Strips, n	$\Theta_{n}$ Simulated by FDTD	$\Theta_{n}$ Calculated by Equation (Equation 1)	Results
3	>90	>90	
3.5	62.3	62.33	Figure 3c
4	50.77	50.80	Figure 2d
4.5	-	43.54	
5	38.3	38.32	Figure 3c
5.5	-	34.30	
6	31.10	31.10	Figure 2d
6.5	-	28.48	
7	-	26.28	
7.5	-	24.41	
8	22.8	22.79	Figure 2d

## Data Availability

The data presented in this study are available upon reasonable request from the corresponding author.

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
