# Peer review of "Double-Resolved Beam Steering by Metagrating-Based Tamm Plasmon Polariton"

_materials, 2022, doi:10.3390/ma15176014_

Round 1
Reviewer 1 Report
Dear Editor,
In the manuscript entitled “Double-resolved beam steering by metasurface based Tamm plasmon polariton” the authors propose a structure composed of a semiconducting material (ITO) sandwiched between a metallic (Ag) diffraction grating and a Bragg reflector. This setup concentrates light in the ITO region. By electrically changing the permittivity below each Ag sheet (separately), they create a phase gradient between neighboring Ag sheets causing the light to bend. However, the crucial part is that this bending couples to a particular diffraction channel that achieves high directivity of the beam.
The paper is original and interesting, however, I have some important remarks for the authors.
Usually, thin structures with the ability to steer light to a particular diffraction order are termed metagratings [https://doi.org/10.1063/5.0012827, https://doi.org/10.1103/PhysRevLett.119.067404, https://doi.org/10.1021/acs.nanolett.5b02524], in contrast to the metasurfaces where light can bend without being coupled to a diffraction channel. Read the introduction of the 1st reference that gives a detailed view. As a result, I propose a respective change in the title and in the Introduction, involving also references such as the ones mentioned here.
My next question is about the Tamm plasmon polariton (TPP). Usually, it is formed at the interface between the Bragg mirror and the metal. Here, we have two intermediate layers of ITO and Al2O3 (the thickness of which, should also be mentioned in the caption of Fig.1(b)). For zero voltage we have a thick dialectic layer sandwiched between a Bragg mirror and a reflecting grating. This seems more like a Fabry-Perot cavity than a TPP. Probably a field profile (to see where the field concentrates) would solve this question.
The next set of questions is how many bilayers constitute the DBR?
What role plays the graphene (write it explicitly in the text)? Do the optical properties of the graphene play any role in the formation of the optical cavity?
The captions must include more information for the paper to be easy-to-read.
In Fig. 2(a) I suppose you mean “μm” in the x-axis.
Does the permittivity of Ag change because of the voltage?
Furthermore, I am very confused about why the shift in Fig. 2(a) is so small since the permittivity of the cavity’s material (ITO) changes so drastically…
After the authors' reply to my concerns, I will be able to make a decision on the paper.
Author Response
Reviewer 1:
Dear Editor,
In the manuscript entitled “Double-resolved beam steering by metasurface based Tamm plasmon polariton” the authors propose a structure composed of a semiconducting material (ITO) sandwiched between a metallic (Ag) diffraction grating and a Bragg reflector. This setup concentrates light in the ITO region. By electrically changing the permittivity below each Ag sheet (separately), they create a phase gradient between neighboring Ag sheets causing the light to bend. However, the crucial part is that this bending couples to a particular diffraction channel that achieves high directivity of the beam.
The paper is original and interesting, however, I have some important remarks for the authors.
Comment 1:
Usually, thin structures with the ability to steer light to a particular diffraction order are termed metagratings [https://doi.org/10.1063/5.0012827, https://doi.org/10.1103/PhysRevLett.119.067404, https://doi.org/10.1021/acs.nanolett.5b02524], in contrast to the metasurfaces where light can bend without being coupled to a diffraction channel. Read the introduction of the 1st reference that gives a detailed view. As a result, I propose a respective change in the title and in the Introduction, involving also references such as the ones mentioned here.
Response: The term metasurface has been replaced by metagrating. Corresponding explanation about this term and new references have been added to the text:
Structures in which meta-atoms or metamolecules form two-dimensional periodic lattices are often called metagratings [ 10, 11]. Such constructions have the ability to direct light into a certain diffraction channel, achieving remarkable efficiency even at very large bending angles [6 ,12].
Comment 2:
My next question is about the Tamm plasmon polariton (TPP). Usually, it is formed at the interface between the Bragg mirror and the metal. Here, we have two intermediate layers of ITO and Al2O3 (the thickness of which, should also be mentioned in the caption of Fig.1(b)). For zero voltage we have a thick dialectic layer sandwiched between a Bragg mirror and a reflecting grating. This seems more like a Fabry-Perot cavity than a TPP. Probably a field profile (to see where the field concentrates) would solve this question.
Response:
We thank the Reviewer for this comment. The ITO and Al2O3 layer thicknesses have been added to the caption of figure 1. It should be noted that the structure in the inset of Figure 1b is presented schematically. The ITO and Al2O3 layers are stretched to demonstrate the nonuniform distribution of charges in the ITO layer. The total thickness of the two layers (ITO+Al2O3) is 25 nanometers, so the observed resonance is a Tamm plasmon polariton, and not a Fabry-Perot cavity mode. Additional explanations about the schematic representation of the structure inserted in Figure 1b have been added to the caption: The ITO and Al2O3 layer thicknesses are 20 nm and 5nm, respectively. The structure in the inset of Figure 1b is presented schematically to demonstrate the non-uniform distribution of charges in the ITO in the case of applying a bias voltage between Ag nanostripes and monolayer graphene.
Comment 3: The next set of questions is how many bilayers constitute the DBR?
Response: The number of DBR layers has been added to the text: The number of DBR layers is equal to 15
Comment 4: What role plays the graphene (write it explicitly in the text)? Do the optical properties of the graphene play any role in the formation of the optical cavity?
Response: We thank the Reviewer for this comment.
The graphene layer was used as a second contact to apply bias voltage to the ITO layer. In the considered model, we did not take into account the influence of graphene on the optical properties of the observed resonances. Corresponding explanations have been added to the text: The structure in the inset of Figure 1b is presented schematically to demonstrate the non-uniform distribution of charges in the ITO in the case of applying a bias voltage between Ag nanostripes and monolayer graphene.
Comment 5: The captions must include more information for the paper to be easy-to-read.
Response: The captions have been reworked.
Comment 6: In Fig. 2(a) I suppose you mean “μm” in the x-axis.
Response: This inacuracy has been corrected.
Comment 7: Does the permittivity of Ag change because of the voltage?
Response: The change of the Ag dielectric permittivity was not taken into account in our calculations: The change in the dielectric constant of Ag under the bias voltage was not taken into account.
Comment 8: Furthermore, I am very confused about why the shift in Fig. 2(a) is so small since the permittivity of the cavity’s material (ITO) changes so drastically…
Response: Indeed, the shift of the resonant wavelength is insignificant. Such a slight change in wavelength was also observed in the work (Huang, Y.W.; Lee, H.W.H.; Sokhoyan, R.; Pala, R.A.; Thyagarajan, K.; Han, S.; Tsai, D.P.; Atwater, H.A. Gate-Tunable Conducting Oxide Metasurfaces. Nano Letters 2016, 16, 5319–5325). In such a device, the phase of the reflected wave is important, which, in turn, changes by more than pi.
After the authors' reply to my concerns, I will be able to make a decision on the paper.

Reviewer 2 Report
It may be better if the result can be verified by some alternative approach. Moreover, the comparison in the tabular form is highly appreciated to highlight the novelty.
Author Response
Reviewer 2:
It may be better if the result can be verified by some alternative approach. Moreover, the comparison in the tabular form is highly appreciated to highlight the novelty.
Response: The reviewer did not specify which approach they suggest for verification. Аs far as we understood from the reviewer's comment, it is necessary to compare the results obtained by different methods. In this paper we compared numerical data against the analytical approach from Eq. (1). In our view, such a comparison is sufficient to demonstrate the correctness of the physical model behind the beam steering. The comparison between Eq. (1) and our numerical data is now given in Table 1 which shows that the angles of diffraction found by both approaches are in a good agreement.

Reviewer 3 Report
Comments on the manuscript
In this manuscript entitled “Double-resolved beam steering by metasurface based Tamm plasmon polariton,” the authors presented a systematic study on active metasurfaces with dynamic modulation on the amplitude and phase of incident light. Specifically, the dynamic control of the phase is enabled by changing the number of nanostrips and external voltages. Also, a multi-layer configuration supporting Tamm plasmon polariton was used, which has the potential of building lasers. The manuscript, in general, is interesting and their simulation results are incremental to the field of active metasurfaces. Besides, the manuscript is technically sound with well-supported conclusions and assertions. Also, the language is readable and fluent, and the references are appropriate. Overall, the manuscript is concise with strong logical consistency
Thus, this manuscript meets the scope of Materials. My comments and suggestion to the authors are listed below.
1. My major concern is related to the novelty of the implementation of the multi-layer Tamm configuration. Although it has the potential to build active lasers [Symonds, Clémentine, et al. "Confined Tamm plasmon lasers." Nano letters 13.7 (2013): 3179-3184], the wavefront control of the 1st diffraction presented in this manuscript is weakly related to the Tamm configuration. Thus what is the key selling point for the Tamm configuration? Better phase manipulation, fabrication more friendly, or others? The authors need to think seriously about these comments and questions, and briefly highlight the selling point in the abstract since it is closely related to the technical innovations and scientific impact of the manuscript.
2. “Metasurfaces open up new opportunities for implementation of holograms [3], lenses [4], media with anomalous reflection [5,6], lasers [7,8], etc.” Apart from holograms, lenses, and media with anomalous reflection, perfect absorption with critical coupling is an important aspect of metasurfaces, which is missing. [Liang, Yao, et al. "Bound states in the continuum in anisotropic plasmonic metasurfaces." Nano Letters 20.9 (2020): 6351-6356]. Also, ITO film is known for its index near zero feature. Better to include this aspect. [Pang, Kai, et al. "Adiabatic frequency conversion using a time-varying epsilon-near-zero metasurface." Nano Letters 21.14 (2021): 5907-5913].
3. Line 36. Abbreviation issues. “…metal-insulator-metal devices, is the possibility of implementing VCSEL [22], which can…” Better to give the full name before using the abbreviation VCSEL—“vertical-cavity surface-emitting lasers”. Similar situations can be found in other paragraphs. For example “The beam steering in ITO metasurfaces has been…”. “TPPs”.
4. “Nonetheless, half-integer values of N lead to sharp beams, as illustrated in Table 1.” What is the physical principle behind this? More discussion is required.
Author Response
Reviewer 3:
Comments on the manuscript
In this manuscript entitled “Double-resolved beam steering by metasurface based Tamm plasmon polariton,” the authors presented a systematic study on active metasurfaces with dynamic modulation on the amplitude and phase of incident light. Specifically, the dynamic control of the phase is enabled by changing the number of nanostrips and external voltages. Also, a multi-layer configuration supporting Tamm plasmon polariton was used, which has the potential of building lasers. The manuscript, in general, is interesting and their simulation results are incremental to the field of active metasurfaces. Besides, the manuscript is technically sound with well-supported conclusions and assertions. Also, the language is readable and fluent, and the references are appropriate. Overall, the manuscript is concise with strong logical consistency
Thus, this manuscript meets the scope of Materials. My comments and suggestion to the authors are listed below.
Commnt 1: My major concern is related to the novelty of the implementation of the multi-layer Tamm configuration. Although it has the potential to build active lasers [Symonds, Clémentine, et al. "Confined Tamm plasmon lasers." Nano letters 13.7 (2013): 3179-3184], the wavefront control of the 1st diffraction presented in this manuscript is weakly related to the Tamm configuration. Thus what is the key selling point for the Tamm configuration? Better phase manipulation, fabrication more friendly, or others? The authors need to think seriously about these comments and questions, and briefly highlight the selling point in the abstract since it is closely related to the technical innovations and scientific impact of the manuscript.
Response:
Additional clarification has been added to the introduction: The advantage of the proposed TTP-based structure compared to conventional metal-insulator-metal devices is the excitation of resonance with higher Q-factor which makes the phase more sensitive to the control parameters. Moreover, the proposed structure is attractive for implementing a photonic crystal surface-emitting lasers (PCSEL) [28]. Thus, the set-up proposed paves a way for engineering optical devices with both PCSEL and beam steering functionality.
Additional sentence has been added to the abstract: The possibility of doubling the angular resolution of beam steering by using asymmetric reflected phase distribution with integer and half-integer periods of the metagrating is demonstrated.
Comment 2: “Metasurfaces open up new opportunities for implementation of holograms [3], lenses [4], media with anomalous reflection [5,6], lasers [7,8], etc.” Apart from holograms, lenses, and media with anomalous reflection, perfect absorption with critical coupling is an important aspect of metasurfaces, which is missing. [Liang, Yao, et al. "Bound states in the continuum in anisotropic plasmonic metasurfaces." Nano Letters 20.9 (2020): 6351-6356]. Also, ITO film is known for its index near zero feature. Better to include this aspect. [Pang, Kai, et al. "Adiabatic frequency conversion using a time-varying epsilon-near-zero metasurface." Nano Letters 21.14 (2021): 5907-5913].
Response: We thank the reviewer for the interesting references we have not been aware of. The references have been added to the text.
“Metasurfaces open up new opportunities for implementation of holograms [ 3], lenses [4], media with anomalous reflection [5 ,6], lasers [7, 8], perfect absorber with critical coupling [ 9] etc”
“The beam steering in indium tin oxide (ITO) [26 ,27 ]”
Commnt 3: Line 36. Abbreviation issues. “…metal-insulator-metal devices, is the possibility of implementing VCSEL [22], which can…” Better to give the full name before using the abbreviation VCSEL—“vertical-cavity surface-emitting lasers”. Similar situations can be found in other paragraphs. For example “The beam steering in ITO metasurfaces has been…”. “TPPs”.
Response: These inaccuracies have been corrected.
“supporting Tamm plasmon polaritons (TTP) [21 –25]. The beam steering in indium tin oxide (ITO) [26 ,27 ]”
“ is attractive to implementing photonic crystal surface-emitting laser (PCSEL)”
Commnt 4: “Nonetheless, half-integer values of N lead to sharp beams, as illustrated in Table 1.” What is the physical principle behind this? More discussion is required.
Response: We thank the Reviewer for this comment. This confusing passage has been reworked:
In principle, n can be an arbitrary real number providing continuous steering, rather than discrete steering for integer n. The drawback of non-integer n is that the TPPs of neighboring strips are mixed, their phases produce unrepeatable cell patterns that are hard to control. Nevertheless, rational n produces few different patterns that are periodic in space. For example, the half-integer value of n makes only two different cell patterns that lead to relatively sharp beams, as illustrated in Table 1. If three and more patterns are included, then the +-1 diffraction order intensity falls down and requires special inverse design of applied voltages for effective beam steering.

Round 2
Reviewer 1 Report
Dear Editor,
The authors have answered my questions and taking into account the novelty of this work I, suggest publication in your Journal.
Reviewer 3 Report
The authors have addressed my comments and improved the manuscript substantially. The manuscript is scientifically sound. Thus, I have no further questions but to give my proposal of acceptance. Good luck to the authors.